# A systematic scoping review of health-promoting interventions for contact centre employees examined through a behaviour change wheel lens

Zoe Bell[1], Lorna Porcellato[2], Paula Holland[3], Abigail Morris[3], Chloe Smith[2], Charlotte Haines[2], Lee Graves[1]*

1 Research Institute for Sport and Exercise Sciences, Liverpool John Moores University, Liverpool, United Kingdom, 2 Public Health Institute, Liverpool John Moores University, Liverpool, United Kingdom, 3 Division of Health Research, Lancaster University, Lancaster, Lancashire, United Kingdom

* L.E.Graves@ljmu.ac.uk

**Data Availability Statement:** All relevant data are within the paper and its Supporting information files.

## Abstract

### Purpose

Social determinants of health and poor working conditions contribute to excessive sickness absence and attrition in contact centre advisors. With no recent review conducted, the current scoping review is needed to investigate the volume, effectiveness, acceptability, and feasibility of health-promoting interventions for contact centre advisors. This will inform the adoption and implementation of evidence-based practice, and future research.

### Methods

Searches conducted across four databases (MEDLINE, PsycInfo, CINAHL, Web of Science) and reference checking in February 2023 identified health-promoting interventions for contact centre advisors. Extracted and coded data from eligible interventions were systematically synthesised using the nine intervention functions of the Behaviour Change Wheel and behaviour change technique taxonomy.

### Results

This scoping review identified a low number of high quality and peer-reviewed health-promoting intervention studies for contact centre advisors (28 studies since 2002). Most interventions were conducted in high-income countries with office-based advisors, predominantly using environmental restructuring and training strategies to improve health. Most interventions reported positive effectiveness results for the primary intended outcomes, which were broadly organised into: i) health behaviours (sedentary behaviour, physical activity, smoking); ii) physical health outcomes (musculoskeletal health, visual health, vocal health, sick building syndrome); iii) mental health outcomes (stress, job control, job satisfaction, wellbeing). Few interventions evaluated acceptability and feasibility.

**Funding:** ZB received funding from the National Institute for Health Research Applied Research Collaboration North West Coast, UK (ARC NWC) (Grant number: 293). The views expressed in this publication are those of the author(s) and not necessarily those of the National Institute for Health Research or the Department of Health and Social Care, UK. https://arc-nwc.nihr.ac.uk/. The funders had no role in study design, data collection and analysis, decision to publish, or preparation of the manuscript.

**Competing interests:** The authors have declared that no competing interests exist.

## Conclusion

There is little evidence on the effectiveness, acceptability, and feasibility of health-promoting interventions for contact centre advisors. Evidence is especially needed in low-to-middle income countries, and for remote/hybrid, nightshift, older and disabled advisors.

## Introduction

It is estimated that over 4% of the UK's working population is employed in a contact centre [1]. Contact centre advisors handle customer queries through multiple platforms (phone calls, chat/messaging, email) and help enhance an organisation's image [2]. Within this role, advisors typically experience verbal aggression from customers [3], repetitive tasks, fixed breaks, low autonomy [4, 5] and continuous performance monitoring [6] in a noisy [7] and sedentary [8] environment. These working conditions contribute to visual, auditory and vocal fatigue, psychological distress, musculoskeletal discomfort [9], and increased risk of developing non-communicable diseases and premature mortality [10]. Advisors typically receive low pay and have low levels of education [11, 12]. These social determinants of health are associated with engagement in unhealthy lifestyle behaviours (low physical activity [13], poor diet [14], smoking [15], higher alcohol consumption [16]). These determinants combine with the aforementioned poor working conditions to contribute to higher rates of sickness absence (3.7% [17] vs 1.9% [18]) and attrition, the pace at which people leave the company, (21% [19] vs 15% [20]) in contact centre advisors compared to UK averages across all industries. Accordingly, contact centres are a priority setting for health promotion to reduce health inequalities and the economic burden of absenteeism and attrition.

Trade (labour) unions and private sector organisations have produced strategy and guidance documents [21, 22] to support contact centres to adopt and implement health-promoting regulations and solutions for employees [23]. The health and wellbeing solutions within these documents however are not (or not transparently) evidence-informed, and appear based on expert advice, which may be biased [24]. The promotion of evidence-informed solutions/interventions to contact centres is important for facilitating (cost) effective regulation, practice and sustained positive change [25], however little is known regarding health-promoting interventions for contact centre advisors.

Only one non-peer reviewed publication has examined the effectiveness of interventions to improve the health, wellbeing and/or performance of contact centre employees [26]. Sixteen intervention studies were identified relating to ergonomic conditions, job redesign, air quality, stress reduction and vocal training, however, four studies did not assess health or wellbeing outcomes, and searches were up to July 2010. This highlights the need for an up-to-date review of health-promoting interventions for contact centre employees (especially advisors) to inform the development of health strategy and guidance documents for contact centres and aid the planning and commissioning of future research.

This scoping review examined the evidence for health-promoting interventions for contact centre employees and addresses four research questions:

1. What is the extent, range, nature, and quality of the intervention evidence?

2. What is the current evidence regarding intervention effectiveness?

3. What is the current evidence regarding intervention acceptability and feasibility?

4. What are the evidence gaps requiring further research?

## Methodology

This scoping review was conducted according to the Joanna Briggs Institute (JBI) methodology for scoping reviews [27–29]. The review was preregistered on the *Open Science Framework* on the 12<sup>th</sup> April 2022 [30] and is reported in accordance with the PRISMA extension for scoping reviews [31] (see S1 File for PRIMSA-ScR checklist).

### Search strategy

The search strategy located published studies. One researcher (ZB) searched MEDLINE, PsycInfo, CINAHL, Web of Science (S2 File: search strategies) and Google Scholar databases on the 21<sup>st</sup> February 2023. The reference lists of all included sources of evidence were screened for additional studies, alongside relevant citation searches.

### Eligibility criteria

The inclusion criteria for eligible intervention studies (based on behaviour change wheel (BCW) definitions; see explanation in 'synthesis of results' below [23]) were: (a) directly or indirectly related to improving the health of contact centre employees; (b) published in English; (c) published since 2002. Studies published prior to 2002 were excluded as a previous review [26] identified no relevant research before this.

### Evidence selection

Identified citations were collated and uploaded into Endnote (Version X9) with duplicates removed using Endnote's duplicate identification strategy and then manually. References were uploaded to the screening tool Rayyan [32] for independent assessment by two reviewers (ZB, CS) against inclusion criteria. The same two reviewers independently screened all titles and abstracts, followed by full-text assessments for eligible citations. Any disagreements between reviewers were resolved through discussion with an additional author (LG).

### Charting the data

Two reviewers (ZB, CS) developed, tested and calibrated a data-charting tool in Excel by extracting data from four randomly selected documents. Discussions of the results informed tool adaptations. For the full data-charting process, each source was charted independently by two reviewers (ZB, CS). Data was collated with any disagreements resolved through discussion.

**Data Items.**  To address research question one, data were extracted on intervention characteristics (citation details, place published, country of origin), aim, and methodological characteristics (participant and contact centre details, study design, intervention delivery), and underpinning theories. Author conclusions for each intervention were extracted to address research question two (effectiveness) and three (acceptability and feasibility). The acceptability of interventions was explored by the authors of the papers using qualitative methods, with studies reporting perceived experiences of the interventions. The final charting form (S3 File) presents clear definitions of each data item.

### Critical appraisal of individual sources of evidence

We critically appraised the quality of included interventions by assessing the risk of bias that each study displays. This appraisal did not impact the inclusion decisions, as guided by a scoping review framework [28]. We used the Cochrane RoB2 tool [33] to appraise randomised controlled trials, the ROBINS-I tool [34] to appraise quasi-experimental trials and the NHLBI

quality assessment tool for pre-post studies [35]. One pre-post study was not appraised, as the main focus of the study was to assess the acceptability and feasibility of the pre-post trial (see S4 File for included studies reference list:15).

## Synthesis of results using the Behaviour Change Wheel (BCW)

Sources identified were mapped to the nine intervention functions of the BCW (education, enablement, training, coercion, restriction, environmental restructuring, incentivisation, persuasion, modelling) [23] to systematically describe each intervention, and the behaviour change techniques (BCT) used [36]. A detailed account of the BCW is available [37]. This approach will support researchers and stakeholders to interpret the evidence-base, informing future research and practice. To address research question one, extracted characteristics summarise the extent, range and nature of the evidence. Within this, two reviewers (ZB, CH) systematically coded intervention components within included studies to a) the nine BCW intervention functions, and b) 93 BCT [36] using detailed intervention descriptions (S1 Table: intervention description table). One reviewer (ZB) had completed BCT taxonomy training. Results were synthesised using relational analysis to present the interventions by their main intended outcomes; this method allows for a rich 'joined-up description' within the analysis [38]. Accordingly, we present findings for research question 2 (effectiveness) and three (acceptability and feasibility) interchangeably within the results. Evidence gaps are discussed throughout to address research question four.

## Results

### Selection of sources of evidence

A PRISMA study flow diagram [39] (Fig 1) details the screening process and reasons for exclusion at full text. Database searches and reference checking returned 328 records. After removing duplicates, 231 titles and abstracts were screened, and the full text of 40 records were screened. Fourteen records were excluded resulting in 26 eligible records for research question one. Two articles (S4 File:10,22) reported two separate and eligible intervention studies. Accordingly, 28 intervention studies from 26 intervention articles were eligible for research question two (intervention effectiveness). Five intervention studies were eligible for research question three (intervention acceptability and feasibility). A detailed description of each intervention is available (S1 Table).

### Characteristics of sources of evidence

Related to research question one, 14 studies were published between 2003–2011 and 14 between 2012–2022. Most of the 28 intervention studies were conducted in high-income countries (S2 Table: characteristics of included intervention studies): USA (6/28, 21.4%), UK (5/28, 17.9%), Australia (4/28, 14.3%), Germany (2/28, 7.1%) and one each (3.6%) in Finland, Austria, Denmark, Singapore and Taiwan China. Five interventions were conducted in upper middle-income countries (South Africa, 3/28, 10.7%; Turkey, 1/28, 3.6%, Iran, 1/28, 3.6%) and one intervention in a lower middle-income country (India, 1/28, 3.6%). No studies were conducted in low-income countries. The number of participants totalled 2,774 with samples ranging from 14 (S4 File:11,12) to 646 (S4 File:14). Most studies included contact centre advisors only (23/28, 82.1%). One study each (3.6%) recruited advisors with a disability (S4 File:4), voice problems (S4 File:5), employees who smoke (including advisors, managers, admin staff, researchers/analysts) (S4 File:14), advisors and team leaders (S4 File:16), and all employees (including advisors, admin staff, support staff) (S4 File:7). From studies reporting participant

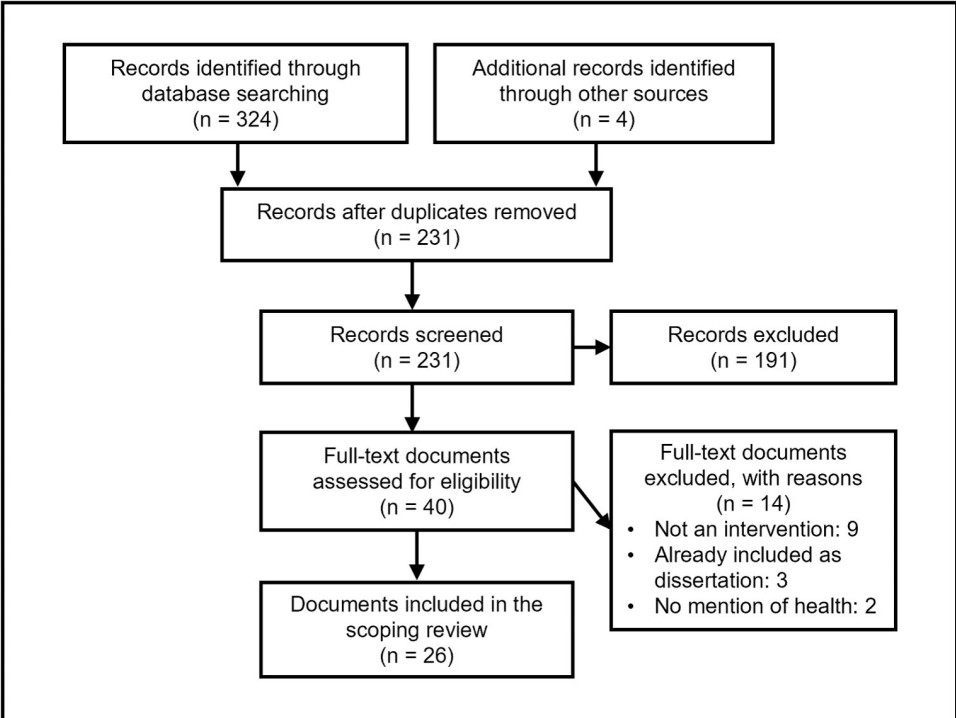

**Fig 1. PRISMA scoping review flow diagram.**

age (19/28, 67.9%), the mean was 32.5 years (mean range 23.1 (S4 File:14) to 40.0 years (S4 File:1,18)). From studies reporting participant gender (25/28, 89.3%), the mean proportion of females was 65.7% (range 19.7% (S4 File:14) to 100% (S4 File:21,26)) and males was 34.3% (range 0% (S4 File:21,26) to 80.3% (S4 File:14)). From studies reporting participant ethnicity (6/28, 22.2%), Caucasian was most represented (mean 77.7%, range 47.8% (S4 File:18) to 100% (S4 File:12)).

Ten of the 28 studies (35.7%) were randomised controlled trials (RCTs) (including two clustered RCTs), eight (28.6%) were quasi-experimental trials (controlled before and after), and ten (35.7%) were pre-post studies (within-subjects design). Five interventions were single component (5/28, 17.9%) (S4 File:4,21,22,23; note, S4 File:22 reported two separate and eligible intervention studies within one article). The remainder were multicomponent (23/28, 82.1%).

In relation to the BCW, environmental restructuring was used in 24/28 (85.7%) intervention studies, followed by training (19/28, 67.9%), education (12/28, 42.9%), enablement (10/28, 35.7%), persuasion (6/28, 21.4%), incentivisation (2/28, 7.1%), and modelling (1/28, 3.6%). No study used coercion or restrictions. The three most used BCT were *instruction on how to perform the behaviour* (training function), *adding objects to the environment* (environmental restructuring function) and *behavioural practice and rehearsal* (training function). See 'Synthesis of evidence by intervention outcome' section for full BCT details.

Twelve of the 28 (42.9%) studies were underpinned by theory, including stress/mindfulness theory (5/28, 17.9%), job redesign theory (5/28, 17.9%) and behaviour change theory/the socioecological model (2/28, 7.1%). Nine interventions lasted <3 months (32.1%), ten lasted 3–6 months (35.7%) and five >6–12 months (17.9%). Intervention length was unclear for four studies (14.3%). Most interventions occurred in an office setting and one of these interventions

included a home-based component (S4 File:1). The intervention delivery/implementation location was unclear in two studies (S4 File:14,26). Over half the interventions involved researchers delivering all or part of the intervention (15/28, 53.6%). This was followed by interventions partly delivered by individuals working within the organisations (participatory research participants, team leaders, management; 5/28, 17.9%). One study each (3.6%) had all, or part of the intervention delivered by either group facilitators with previous experience of receiving the intervention, a clinical councilor/social worker, an occupational health and safety officer, a speech teacher/language therapist, an expert tobacco counsellor, or an external consultant in organisational development. It was unclear who delivered the intervention in eight studies (S4 File:4,6,9,14,21,23,24,26).

Many outcomes were measured, including health outcomes in 19/28 intervention studies (67.9%; stress-related indicators, visual fatigue, musculoskeletal discomfort, job related wellbeing, vocal health), behavioural outcomes in 6/28 studies (21.4%; sitting time, physical activity, tobacco use), indirect measures of health in 3/28 studies (10.7%; job control, job satisfaction), and intervention acceptability and/or feasibility in 5/28 studies (17.9%).

## Source quality

For the RCTs, four studies had low bias for all sections, five had some concerns for the measurement of the outcome, and two of these also had high bias for adherence to the intervention (S3 Table: ROB assessment tables). One study had some concerns for assignment to the intervention and the selection of reported results, and another had some concerns with the randomization process. Risk of bias was generally higher for the quasi-experimental studies than the RCTs, typically due to confounding in five of the eight studies (S3 Table). None of these studies received low bias for all categories. Some concerns arose for deviations from the intervention due to poor adherence and for measurement of the outcome due to self-report measurements. One article (S4 File:10) lacked sufficient information to reliably judge the quality of each section. For pre-post studies, six of the included studies were judged to be 'fair' and three were 'poor' in relation to their risk of bias (S3 Table). One study (S4 File:26) did not report receiving ethical approval.

## Synthesis of evidence by intervention outcome

The intervention studies were mapped to the BCW intervention categories and BCT, and synthesised to display the total number of functions used across all interventions (Table 1). The interventions were then categorised into the following sections based on the reported primary outcome or intended primary aim: i) health behaviours (sedentary behaviour, physical activity, smoking); ii) physical health outcomes (musculoskeletal health, vocal health, visual health, sick building syndrome); iii) mental health outcomes (stress, job control, job satisfaction, wellbeing). While we acknowledge that most studies measured multiple outcomes (see S2 Table for all the study outcomes i.e., S4 File:15's primary outcome related to sitting time [health behaviour] but they also measured musculoskeletal outcomes [physical health]), this categorisation approach brings order to the synthesis and allows discussion of research question two and three within the following sections.

## Health behaviours

**Sedentary behaviour and/or physical activity.** Five interventions (S4 File:3,6,15–17) primarily targeted sedentary behaviour reduction and/or physical activity promotion. All five interventions utilised stand-capable desks to reduce sitting time (*environmental restructuring*) and at least one other intervention component from a different BCW intervention function:

**Table 1. Summary of studies mapped to the behaviour change wheel (BCW) intervention functions and behaviour change techniques (BCT).**

| BCW intervention function | BCT* and intervention studies** | Number of studies using the BCT | Number of studies using the intervention function |
|---|---|---|---|
| **Environmental restructuring (change the physical or social context)** | **12.5 Adding objects to the environment**: Sit-stand desk (S4 File:3); Screen filter (S4 File:4); Ergonomic checklist (S4 File:10); A silent room (S4 File:11,12); Height-adjustable workstations (S4 File:15,16); Stand-capable desks (S4 File:6,17); Armband and trackball (S4 File:18); New filter and outdoor air supply (S4 File:23); Voice biofeedback (S4 File:19); Heart rate variability biofeedback (S4 File:9); Office plants (S4 File:22[study 1 and 2]); Adjustable chairs with arm rests, footrests and screen stands (S4 File:20) | 16 | 24 |
| | **12.1 Restructuring the physical environment**: Forearm support (S4 File:5); Filter and outdoor air supply (S4 File:23); Temperature and outdoor air supply (S4 File:21); Modifications made to the physical workstation (S4 File:20) | 4 | |
| | **12.2 Reconstructing the social environment**: Job redesign changes (S4 File:2); Job redesign changes (S4 File:7,8); Alignment job design, high-involvement work processes and autonomous work teams (S4 File:24,25); Given an additional 10-minute rest break to perform exercise program (S4 File:20) | 6 | |
| | **2.6 Biofeedback**: Heart rate variability biofeedback (S4 File:9); Voice biofeedback (S4 File:19) | 2 | |
| **Training (imparting skills)** | **4.1 Instruction on how to perform the behaviour:** Guided meditation (S4 File:1); Sit-stand desk use (S4 File:3) [58]; Forearm positioning (S4 File:5); Skill training to increase job control (S4 File:7,8); Ergonomic checklist and skill-based training programme for MSD (S4 File:10[study 1 and 2]); Progressive muscle relaxation instructions (S4 File:11,12); Vocal training (S4 File:13); Training session on posture changes, active breaks and standing work (S4 File:15,16); Stand-capable desk use (S4 File:6,17); Ergonomics training (S4 File:18); 1-week training seminar in high-involvement work processes and autonomous work teams (S4 File:24,25); Diaphragm breathing training (S4 File:26); ergonomic skills training and regular stretching exercises (S4 File:20) | 19 | 19 |
| | **8.1 Behavioural practice/rehearsal:** Guided meditation practice (S4 File:1); Skill training to increase job control (S4 File:7,8); Skill-based training programme for MSD (S4 File:10[study2]); Progressive muscle relaxation practice (S4 File:11,12); Vocal training (S4 File:13); Training seminar to encourage a participative environment (S4 File:24,24); Diaphragm breathing training (S4 File:26) | 10 | |
| | **6.1 Demonstration of the behaviour:** Skill training to increase job control (S4 File:7,8); Skill-based training programme for MSD (S4 File:10[study2]); Vocal training (S4 File:13); Diaphragm breathing training (S4 File:26); Visual pamphlet on ergonomic skills training (S4 File:20) | 6 | |
| **Education (increase knowledge or understanding)** | **5.1 Information about health consequences**: Educational stress management articles (S4 File:1); Educated on the benefits of MSD prevention training (S4 File:10[study2]); Health hazards of tobacco (S4 File:14); Vocal hygiene (S4 File:13); Education sessions on posture changes, active breaks and standing work (S4 File:15,16); Voice hygiene (S4 File:26); Ergonomic training on the etiology of MSD (S4 File:20) | 8 | 12 |
| | **2.2 Feedback on behaviour**: Heart rate variability biofeedback (S4 File:9); Voice biofeedback (S4 File:19) | 2 | |
| | **2.7 Feedback on outcomes of behaviour**: Feedback on anthropometric, cardiometabolic and behavioural outcomes (S4 File:15) | 1 | |
| | **5.3 Information about the social and environmental consequences**: Lunch and learn sessions in high-involvement work processes (S4 File:24,25) | 2 | |

*(Continued)*

**Table 1.** (Continued)

| BCW intervention function | BCT* and intervention studies** | Number of studies using the BCT | Number of studies using the intervention function |
|---|---|---|---|
| **Enablement [increase means or reduce barriers to increase capability (beyond education or training) or opportunity (beyond environmental restructuring)]** | **1.2 Problem Solving**: Steering group to identify problematic aspects of work organisation to recommend job redesign action (S4 File:2); Assessment to identify problematic aspects of work organisation to recommend job redesign action (S4 File:7,8); Advisors worked collectively to identify practical strategies for moving more (S4 File:15,16); Identifying and adjusting measurement and reward systems in alignment job redesign, team problem solving for job redesign needs in high-involvement work processes and autonomous work teams (S4 File:24,25); Focus groups and one-to-one therapy sessions to address rationalizations for continued tobacco use (S4 File:14); snapshots of inappropriate exercises taken to discuss potential solutions (S4 File:20) | 9 | **10** |
| | **3.1 Social support (unspecified)**: Group discussion and sharing positive experiences (S4 File:1); Mentors assigned in high-involvement work processes (S4 File:24,25); Focus group support (S4 File:14) | 4 | |
| | **1.4 Action planning**: Job redesign actions (S4 File:2); Job redesign actions (S4 File:7,8); Job redesign actions teams (S4 File:24,25) | 5 | |
| | **1.1 Goal setting (behaviour)**: Goal setting to increase standing and walking (S4 File:15,16) | 2 | |
| | **1.5 Review behaviour goal(s)**: Participants meet to review job redesign goals (S4 File:7,8) | 2 | |
| | **1.7 Review outcome goal(s)**: Participants meet to review job redesign goals (S4 File:7,8) | 2 | |
| | **2.4 Self-monitoring of outcome(s) of behaviour**: Participants monitor outcomes of job redesign changes (S4 File:7,8); Team measures own performance in autonomous work teams (S4 File:24,25) | 4 | |
| | **2.3 Self-monitoring of behaviour**: Daily standing and walking time (S4 File:15); Log given to track daily exercises performed (S4 File:20) | 2 | |
| | **11.1 Pharmacological support**: Pharmacotherapy for smoking cessation (S4 File:14) | 1 | |
| | **11.2 Reduce negative emotions**: Pharmacotherapy for smoking cessation (S4 File:14) | 1 | |
| | **1.8 Behavioural contract**: Written agreements of tasks and roles (S4 File:24,25) | 2 | |
| | **2.1 Monitoring of behaviour by others without feedback:** Researchers monitored ergonomic behaviours and participation in the regular exercise program (S4 File:20) | 1 | |
| **Persuasion (use communication to induce positive or negative feelings to stimulate action)** | **9.1 Credible source**: Stand-up champions and team leaders (S4 File:15,16); Expert tobacco counsellor (S4 File:14); Clinical support (S4 File:1) | 4 | **6** |
| | **7.1 Prompts/cues**: Email reminders to practice mindfulness (S4 File:1); Daily email reminders to stand (S4 File:3); Email reminders to stand (S4 File:15,16) | 4 | |
| | **10.10 Reward (outcome)**: Points awarded for smooth waves (S4 File:9) | 1 | |
| **Incentivisation (create an expectation of reward)** | **2.2 Feedback on behaviour**: Positive feedback for aligned behaviours in alignment job redesign (S4 File:24,25) | 2 | **2** |
| | **2.7 Feedback on outcome(s) of behaviour:** Positive feedback for achieving aligned goals in alignment job redesign (S4 File:24,25) | 2 | |
| | **10.4 Social reward:** Expressions of management approval in alignment job redesign (S4 File:24,25) | 2 | |
| | **10.2 Material reward (behaviour):** Bonuses and raises in alignment job redesign and merit increases in autonomous work teams (S4 File:24,25) | 2 | |

*(Continued)*

**Table 1.** (*Continued*)

| BCW intervention function | BCT* and intervention studies** | Number of studies using the BCT | Number of studies using the intervention function |
|---|---|---|---|
| **Modelling (provide an example for people to aspire to emulate)** | **6.1 Demonstration of the behaviour:** Stand-up champions model standing behaviours (S4 File:15) | 1 | **1** |

MSD: Musculoskeletal Disorder.

S: Supplementary.

*The BCT taxonomy organizes the 93 techniques into a cluster of 16 groups. The table reports the category and technique numbers, i.e. '12.5 Adding objects to the environment' is the 5th technique within the 12th category named 'antecedents'.

**See S4 File for intervention study reference list.

*education* (S4 File:15,16), *persuasion* (S4 File:3,15,16), *training* (S4 File:3,6,15–17), *modelling* (S4 File:15) and *enablement* (S4 File:15,16). Positive effects were most reported for sitting time and standing time outcomes compared to physical activity outcomes. Stand-capable desks increased productivity (S4 File:6), however one study (S4 File:16) stated that stand-capable hot desks were not perceived by participants as feasible. Overall, interventions were accepted (S4 File:15,16), with participants perceiving increased comfort as a factor influencing their standing time (S4 File:17).

**Smoking cessation.** One intervention aimed to encourage smoking cessation (S4 File:14) using three variations of the intervention. The first intervention arm included a health education session followed by an interactive focus group, the second arm additionally included one-to-one behavioural therapy, and the third arm further included pharmacotherapy. Each intervention arm was mapped to varying BCT within *education*, *enablement* and *persuasion*. Each intervention arm increased smoking quit rates (20%, 19%, 20% respectively) and the reduction in tobacco use was higher when introducing pharmacotherapy (26%, 28%, 46% respectively). Many participants complained of high irritability, though it is not clear in the study what this irritability related to.

## Physical health outcomes

**Musculoskeletal disorders (MSD).** Five interventions (S4 File:5,10,18,20; note, S4 File:10 reported two separate and eligible intervention studies within one article) primarily aimed to reduce musculoskeletal-related discomfort or pain. Four interventions (S4 File:5,10[study 1],18,20) provided and/or adjusted the workstation (*environmental restructuring)*. All interventions featured a component of ergonomic *training*, whilst two interventions (S4 File:10 [study 2],20) also implemented an *educational* component. One intervention also utilised enablement (S4 File:20). Most interventions reported reductions in pain or discomfort (S4 File:5,10[study 2],18,20) except for one study in which participants found an ergonomic checklist confusing and lengthy (S4 File:10[study 1]).

**Vocal health.** Three interventions primarily aimed to reduce vocal symptoms (S4 File:13), improve vocal health (S4 File:26) or improve vocal performance (S4 File:19). Interventions included a 2-day vocal training course (S4 File:13), voice therapy (S4 File:26) and a biofeedback software (S4 File:19). All interventions *educated* participants on improving vocal hygiene (habits to support a healthy voice), whilst two interventions also provided vocal *training* (S4 File:13,26) and another featured *environmental restructuring* (S4 File:19). All interventions were reported effective after 3–4 weeks of intervention. The perceived experience of short vocal training course (an indicator for acceptability) was reported to be positive overall (S4 File:13).

**Visual health.**  One intervention aiming to reduce visual fatigue (S4 File:4) used *environmental restructuring* by fitting a screen filter on each computer. No beneficial effects were reported at 5 months follow-up.

**Sick building syndrome.**  Two interventions primarily aimed to reduce sick building syndrome symptoms (intensity of dryness symptoms, eyes aching and nose-related symptoms). One study (S4 File:23) measured the interactive effects of a used or new air filter with higher or lower outdoor air support, resulting in four variations of the intervention. Similarly, another study (S4 File:21) measured the interactive effects of higher or lower temperatures with higher or lower outdoor air support, also resulting in four variations. All interventions utilised *environmental restructuring*. The first study (S4 File:23) found that increasing the outdoor air supply rates with new air filters, and replacing used filters with new ones at the high outdoor air supply rate were effective. The second study (S4 File:21) found that increasing outdoor air supply rates at a higher temperature led to a decrease in a cluster of sick building syndrome symptoms.

## Mental health outcomes

**Stress.**  Four intervention studies primarily aimed to reduce stress or stress-related symptoms. Two interventions used a progressive muscle relaxation intervention within a breaktime 'silent room' (S4 File:11,12). One intervention used a heart rate variability biofeedback device to synchronise respiration and heart rate (S4 File:9). Both interventions utilised *environmental restructuring* and *training*, whilst the biofeedback device also used *incentivisation*. Finally, one study investigated three variations of an intervention using an online mindfulness stress management programme (S4 File:1). Each arm featured the web-based programme, with the second and third arms additionally including a group or clinical support to increase adherence, respectively. These arms map to *education*, *persuasion* and *training* intervention functions, and the group and clinical support maps to *enablement*. Each variation of the online mindfulness stress management programme intervention reported positive reductions in stress outcomes. The addition of group support further reduced stress, though the clinical support provided no additional benefits. The progressive muscle relaxation intervention was reportedly effective, especially post-lunchtime, in reducing emotional and motivational strain states (S4 File:11) and cortisol levels (S4 File:12). The biofeedback device was effective for reducing personal stressors (burnout, fatigue, gastrointestinal, headaches). The online mindfulness programme also measured programme feedback, providing insight into intervention acceptability and feasibility. Whilst acceptance was relatively high, researchers identified the lack of time to practice as a potential barrier for successful implementation (S4 File:1).

**Job control and job satisfaction.**  The primary outcome/aim of three intervention studies was to improve job control (S4 File:2) or job satisfaction (S4 File:24,25). All were job redesign interventions, involving *environmental restructuring* and *enablement*. Two studies investigated three variations of job redesign (S4 File:24,25): i) alignment job redesign, ii) high-involvement work processes, and iii) autonomous work teams. Alignment job redesign and autonomous work teams included *incentivisation*, high-involvement work processes included *education* and the latter two included *training*. Most interventions were reported to be effective at increasing job control (S4 File:2) or job satisfaction (S4 File:24,25), except for the autonomous work teams variation.

**Wellbeing.**  Four intervention studies primarily aimed to improve wellbeing (S4 File:7,8,22; note, S4 File:22 reported two separate and eligible intervention studies within one article). Two interventions used participatory job redesign (S4 File:7,8) and two introduced plants to the workplace (S4 File:22[study 1 and 2]). All interventions used *environmental*

*restructuring* for either the social (S4 File:7,8) or physical environment (S4 File:22[study 1 and 2]). Additionally, the job redesign intervention utilised *enablement* and *training*. Both job redesign interventions were reported to be effective, whilst neither of the plant studies improved wellbeing.

## Discussion

### Research question one—What is the extent, range, nature, and quality of the intervention evidence?

This scoping review identified a low number of peer-reviewed, health-promoting intervention studies for contact centre advisors (28 studies since 2002). Comparatively, another review [40] identified 34 studies (2009–2017) for interventions involving sit-stand desks within a traditional office workplace. Given contact centre advisors are at high risk of poor health due to their working conditions [3, 6–8] and social determinants of health [11, 12, 41], there is an urgent need for more health interventions research in this setting.

Globally, the US holds the largest proportion of contact centres, followed by the Philippines and India [42]. Our review highlighted that interventions were mainly conducted in high-income countries (e.g., US), with few conducted in middle- (e.g., Philippines, India) and low-income countries. Contact centre advisors in low-to-middle income countries likely face even greater risks to health (lower pay, lower levels of education, poor housing, poor working conditions [43]) compared to those in higher-income countries. Accordingly, while more intervention research is needed globally, there is a particular need for health intervention research in low-to-middle income countries that employ a large proportion of global contact centre workforce.

Most participants within the intervention studies were relatively young contact centre advisors (mean of 32.5 years) working day shift hours. Only one study focused on disabled advisors and one on advisors with voice problems. Therefore, contact centre advisors underrepresented in the current evidence include older adults, night workers, and disabled workers. This is problematic as night workers are likely to suffer from additional negative effects on sleep quality, food habits, addictions, social and mental health [44], poor working conditions are likely to have a more severe impact on disabled workers, and, amidst an aging population, the highest incidence of mental health short-term disability claims within the work environment are among those aged 40–49 years [45]. Future intervention research that examines the needs of, and develops interventions for, these especially vulnerable contact centre advisor sub-groups, is warranted.

Few studies adopted an RCT design (35.7%, including two clustered RCTs). This number is low compared to 55.9% of RCT's identified within a similar review assessing interventions for reducing sitting at work [40]. Fewer RCT's indicates lower quality evidence to inform intervention guidance. Despite this, it is acknowledged that RCTs pose a high risk of contamination between groups, meaning future research should consider clustered RCT's as a more feasible design within the contact centre setting [11].

The most common intervention functions examined in contact centres were *environmental restructuring* (adding objects to the environment) and *training* (instruction on how to perform the behaviour). Environmental restructuring may be common due to the need to tackle health problems associated with working for prolonged periods on a computer in a static, seated posture [46]. Training may also be common due to established, existing training structures operating within contact centres for employees. In contrast, modelling and incentivisation were seldom used. The *modelling* function was only used in one intervention study (S4 File:15) with stand-up champions encouraging advisors to sit less and move more at work. This was

perceived ineffective, as advisors were often unsure who the champions were. Future interventions using modelling in contact centres should promote awareness of the champions, and may find the effective use of movement champions in non-contact centre office environments informative [47]. Regarding *incentivisation*, only one job redesign intervention (across two studies) aimed to change behaviours through measurement and reward structures (bonuses, raises, management approval). This may be because job redesign interventions require organisational commitment to adjust structural components, or the financial cost of incentives is too high for centres. Health interventions within non-contact centre office environments have effectively used financial incentives to increase employee health [48], which may be informative for future interventions using this method in contact centres. Finally, no interventions featured coercion or restrictions, which have previously been perceived as unacceptable strategies within a workplace environment [49].

Less than half of the interventions identified were underpinned by theory and those without an underpinning theory were mostly ergonomic interventions to improve vocal, visual or musculoskeletal health. This is consistent with previous reviews describing a 'strikingly small' proportion of ergonomic intervention studies with underpinning theory [50], despite researchers identifying relevant theories [51]. Theory may help to explain the mechanisms behind the effect of an intervention, however, research has indicated that theory-based versus no-stated theory interventions do not differ in effectiveness [52]. Theory can be a valuable resource, but it does not always ensure the effectiveness of interventions; theory may be inconsistently operationalised (put into practice), inappropriate for specific contexts or flawed [53, 54].

Few interventions were implemented long-term, with the longest being 1-year. No interventions had follow-up data collection points beyond 1-year, which is similarly reported in another workplace health intervention review [40]. Most interventions were office-based, with only one containing a home-based component (S4 File:1). This is problematic, as the COVID-19 pandemic sparked a shift to hybrid working, with 64% of contact centre advisors working remotely in 2021 and this predicted to continue in the long-term [55]. Accordingly, there is an urgent need for contact centres and researchers to understand the needs of hybrid/remote workers when developing, adopting and implementing health-promoting interventions. More long-term follow-up intervention studies are also needed.

The multiple outcomes evaluated across the identified interventions may be a result of the many behavioural and health issues that contact centre advisors face. However, despite being linked to work-related stress [56] and social determinants of health [13–16], only five intervention studies targeted physical activity/sedentary behaviour, and only one study targeted smoking. Further, no intervention targeted alcohol consumption or diet. This demonstrates a gap in the evidence compared to workplace health interventions targeting diet (17 identified) [57] and alcohol consumption (18 identified) in traditional office environments [58]. Future research may explore whether behavioural interventions reported as effective in more traditional office environments, are equally effective for contact centre employees.

### Research question two—What is the current evidence regarding intervention effectiveness?

Most interventions reported positive effectiveness results for the primary intended outcome. Only four interventions failed to report effective results, including an ergonomic checklist (S4 File:10[study 2]), a screen filter to reduce visual fatigue (S4 File:4) and two studies putting plants into the workplace to improve wellbeing (S4 File:22[study 1 and 2]). These studies can be interpreted as being amongst the most simplistic interventions, based on the BCW intervention function mapping, with the latter three being single component interventions. This is

in-line with a systematic review assessing workplace health promoting interventions which stated that multi-component interventions were more effective than the single-component interventions [59].

Four (14.3%) interventions identified in this review are cited within health strategy and guidance documents for contact centres, as produced by trade (labour) unions and private sector organisations [21, 22, 60]. These interventions focused on air quality and ergonomic training solutions. In contrast, to the authors' knowledge, the remaining 24 intervention studies identified in this review are not cited in any health strategy or guidance document for contact centres. This highlights a lack of translation of published scientific evidence into practice, and the need for better collaboration between researchers and stakeholders concerned with health promotion in contact centres. Further, there is a need for evaluation of the 'good practice' recommendations within existing documents to understand their effectiveness, acceptability, and feasibility. In combination, these actions can help produce evidence-informed health strategy and guidance documents, and promotion of those documents at scale across the industry could improve the health of hundreds of thousands of contact centre advisors.

### Research question three—What is the current evidence regarding intervention acceptability and feasibility?

Overall, there was a low proportion of studies reporting acceptability and/or feasibility (5/28 studies). All studies appeared acceptable to participants (S4 File:1,13,15,16,17). Regarding feasibility, one study stated that stand-capable hot desks were not feasible (S4 File:16) and one study highlighted lack of time as a potential barrier as participants needed more time to practice a mindfulness programme (S4 File:1). This is likely to be a common challenge for contact centre interventions, as advisors have little autonomy and flexibility surrounding break times [61]. More acceptability and feasible research is needed within this setting due to its unique working conditions.

### Strengths and limitations

This is the first systematic scoping review on this topic to be submitted for peer-review and provides a needed update on a non-peer reviewed publication in 2010. This review utilised a comprehensive search strategy across four databases and google scholar to identify health-promoting interventions for contact centre advisors. To ensure all relevant studies were captured, the search strategy and inclusion criteria remained broad, ensuring a physical, mental and social health focus. The coding framework was based on the established BCW and BCT to systematically describe the range and nature of the evidence, providing structure to the findings. The risk of bias assessment for applicable studies provides the reader with an overview of the quality of the evidence-base, highlighting common biases such as confounding within quasi-experimental designs. This resulted in a recommendation for future research to consider clustered RCTs as a preferable study design to reduce bias within contact centre research.

This review's restriction to behavioural and health outcomes could be a limitation. Business and productivity-related outcomes could prove informative for contact centre stakeholders and should be considered for future reviews. This review is also limited in its capacity to make recommendations for the effectiveness of individual interventions, instead this scoping review provides a descriptive account of the available evidence [28]. Excluding studies that were not published in English was also a potential limitation, however, this did not affect the findings of the review as only three studies were not available in English, none were interventions and would not have been eligible for inclusion.

## Conclusion

There is a lack of research evidence on health-promoting interventions for contact centre advisors. Most intervention studies were conducted in high-income countries, and in office-based contact centre advisors, with key research gaps in low-to-middle income countries, and remote/hybrid, nightshift, older and disabled workers. Most intervention studies reported evidence of effectiveness for promoting employee health, though few studies explored intervention acceptability and feasibility. The field needs more higher quality intervention studies using RCT designs, longer evaluation periods, and associated acceptability and feasibility evaluations. Finally, this scoping review has identified and synthesised health intervention research for contact centre employees that can inform future policy and practice in this occupational setting.

## Supporting information

**S1 File. PRISMA-ScR checklist.**
(PDF)

**S2 File. Search strategies.**
(PDF)

**S3 File. Data extraction tool.**
(PDF)

**S4 File. Intervention study reference list.**
(PDF)

**S1 Table. Intervention description table.**
(PDF)

**S2 Table. Characteristics table.**
(PDF)

**S3 Table. ROB assessment tables.**
(PDF)

## Author Contributions

**Conceptualization:** Zoe Bell, Lorna Porcellato, Paula Holland, Abigail Morris, Lee Graves.

**Formal analysis:** Zoe Bell, Chloe Smith, Charlotte Haines.

**Funding acquisition:** Zoe Bell, Lorna Porcellato, Paula Holland, Abigail Morris, Lee Graves.

**Investigation:** Zoe Bell, Chloe Smith, Charlotte Haines.

**Methodology:** Zoe Bell, Lorna Porcellato, Paula Holland, Chloe Smith, Charlotte Haines, Lee Graves.

**Project administration:** Zoe Bell.

**Supervision:** Lorna Porcellato, Paula Holland, Abigail Morris, Lee Graves.

**Visualization:** Zoe Bell.

**Writing – original draft:** Zoe Bell.

**Writing – review & editing:** Zoe Bell, Lorna Porcellato, Paula Holland, Abigail Morris, Chloe Smith, Charlotte Haines, Lee Graves.

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
