## [Decision Letter · Decision Letter 0]

9 Aug 2023

PONE-D-23-11194A systematic scoping review of health-promoting interventions for contact centre employees examined through a behaviour change wheel lens.PLOS ONE

Dear Dr. Graves,

Thank you for submitting your manuscript to PLOS ONE. After careful consideration, we feel that it has merit but does not fully meet PLOS ONE’s publication criteria as it currently stands. Therefore, we invite you to submit a revised version of the manuscript that addresses the points raised during the review process.

We look forward to receiving your revised manuscript.

Kind regards,

Farhana Haque, MBBS, MPH, MSc, PhD

Academic Editor

PLOS ONE

Journal Requirements:

Reviewers' comments:

Reviewer's Responses to Questions

**Comments to the Author**

1. Is the manuscript technically sound, and do the data support the conclusions?

Reviewer #1: Yes

Reviewer #2: Yes

2. Has the statistical analysis been performed appropriately and rigorously? 

Reviewer #1: N/A

Reviewer #2: Yes

3. Have the authors made all data underlying the findings in their manuscript fully available?

Reviewer #1: Yes

Reviewer #2: Yes

4. Is the manuscript presented in an intelligible fashion and written in standard English?

Reviewer #1: Yes

Reviewer #2: Yes

5. Review Comments to the Author

Reviewer #1: A systematic scoping review of health-promoting interventions for contact centre employees examined through a behaviour change wheel lens is a relevant topic which uses a robust approach (BCW) to analyse the findings. Below are some suggestions that may further strengthen the manuscript.

Introduction:

Line 40: For the benefit of the reader, please briefly describe what do you mean by the term attrition. Also, figures for contact centre employees are compared with UK averages. Please briefly say who these averages relate to? (e.g. office based employees etc?).

Methodology:

Line 75: Can authors justify why they excluded non-English language studies at the eligibility stage especially when they mention that there was a low number of studies from low-and-middle income countries (LMICs)? Including non-English language studies at an earlier stage would have given an idea of how many were actually there and if required (because of issue of translating them into English or non-expertise in other languages), then they may have been excluded at a later stage. Including only studies published in English should be added as a limitation of the review.

Line 92: Authors mention extracting data on acceptability of the intervention. It would also be beneficial to know how acceptability was explored in the included studies (using qualitative, mixed methods?), and whether the authors considered or not considered a mixed methods review instead of just including study author's conclusion as evidence of acceptability and/or feasibility, and the reasons for doing so.

Line 98: If quality of included studies was assessed then I would encourage authors to look into assessing quality of pre-post study designs as well. Examples of some tools that can be considered are NIH study quality assessment tool for before-after (pre-post) studies with no control group (https://www.nhlbi.nih.gov/health-topics/study-quality-assessment-tools) or Evidence project risk of bias tool (Kennedy, C.E., Fonner, V.A., Armstrong, K.A. et al. The Evidence Project risk of bias tool: assessing study rigor for both randomized and non-randomized intervention studies. Syst Rev 8, 3 (2019). https://doi.org/10.1186/s13643-018-0925-0).

Results:

Line 182: Can authors also say a little about interventions mapping to multiple domains.

Line 217: Please briefly elaborate on the term 'vocal hygiene'

Line 231: "For the used-to-new air filter, participant acceptability of air quality decreased" can this sentence be rephrased, this is not very clear.

Discussion:

Line 316: Briefly say why that is so (e.g. because no stated theory does not mean the intervention is not logical/sensible).

Reviewer #2: Thank you for the opportunity to review this well-written article. It fills a substantial gap in current literature and I thoroughly enjoyed reading it.

Some minor comments for the authors consideration:

L36 – is deprived the correct term? or is it lower socioeconomic background(s)?

L40 – are the UK averages comparable to similar job types (ie desk-based workers) or all industries?

L102 – suggest using BCT throughout, rather than behaviour change taxonomies, and please add reference.

L108 – should this read …had ‘completed’ BCT training… ?

L126-143 – this might work better in tabulated format.

L167-175 – again, tabulate?

L282 – where it says ‘relatively young’ it would be useful to quantify what this means.

L310-311 – could you please add some discussion points to explain why coercion and restriction would be of interest to future research.

Discussion section – i) RQ2 and 3 read like a summary of the findings and would benefit from the addition of discussion points. ii) It would be useful for the reader to have all the RQs repeated as the headers in the discussion section – I felt scrolling back to remind myself what they were was disruptive to reading. iii) Should there also be some commentary for RQ4?

6. PLOS authors have the option to publish the peer review history of their article (what does this mean?). If published, this will include your full peer review and any attached files.

Reviewer #1: No

Reviewer #2: **Yes: **Dr Sarah Morton

---

## [Author Response · Author response to Decision Letter 0]

15 Sep 2023

Editor comments: We have made changes to the formatting to meet the PLOS ONE requirements. Thank you for providing the relevant guidance. 

Reviewer 1: We have made changes to the manuscript based on your comments. Thank you, they were very helpful. 

Reviewer 2: We have made changes based on your comments. Thank you for your suggestions, we found them very helpful.

Our responses to the reviewers and editor comments can be found in the uploaded file named 'Response to Reviewers'.

---

## [Decision Letter · Decision Letter 1]

22 Jan 2024

A systematic scoping review of health-promoting interventions for contact centre employees examined through a behaviour change wheel lens.

PONE-D-23-11194R1

Dear Dr. Graves,

We’re pleased to inform you that your manuscript has been judged scientifically suitable for publication and will be formally accepted for publication once it meets all outstanding technical requirements.

Kind regards,

Farhana Haque, MBBS MPH MSc PhD

Academic Editor

PLOS ONE

---

## [Editor Report · Acceptance letter]

29 Feb 2024

PONE-D-23-11194R1 

PLOS ONE

Dear Dr. Graves, 

I'm pleased to inform you that your manuscript has been deemed suitable for publication in PLOS ONE. Congratulations! Your manuscript is now being handed over to our production team.

Kind regards, 

on behalf of

Dr Farhana Haque 

Academic Editor

PLOS ONE